# Discovery of several thousand highly diverse circular DNA viruses

**Michael J Tisza[1], Diana V Pastrana[1], Nicole L Welch[1], Brittany Stewart[1], Alberto Peretti[1], Gabriel J Starrett[1], Yuk-Ying S Pang[1], Siddharth R Krishnamurthy[2], Patricia A Pesavento[3], David H McDermott[4], Philip M Murphy[4], Jessica L Whited[5,6,7], Bess Miller[5,6], Jason Brenchley[8], Stephan P Rosshart[9], Barbara Rehermann[9], John Doorbar[10], Blake A Ta'ala[11], Olga Pletnikova[12], Juan C Troncoso[12], Susan M Resnick[13], Ben Bolduc[14], Matthew B Sullivan[14,15], Arvind Varsani[16,17], Anca M Segall[18], Christopher B Buck[1]\***

[1]Lab of Cellular Oncology, National Cancer Institute, National Institutes of Health, Bethesda, United States; [2]Metaorganism Immunity Section, Laboratory of Immune System Biology, National Institute of Allergy and Infectious Diseases, National Institutes of Health, Bethesda, United States; [3]Department of Pathology, Microbiology, and Immunology, University of California, Davis, Davis, United States; [4]Molecular Signaling Section, Laboratory of Molecular Immunology, National Institute of Allergy and Infectious Diseases, National Institutes of Health, Bethesda, United States; [5]Department of Orthopedic Surgery, Harvard Medical School, The Harvard Stem Cell Institute, Brigham and Women's Hospital, Boston, United States; [6]Broad Institute of MIT and Harvard, Cambridge, United States; [7]Department of Stem Cell and Regenerative Biology, Harvard University, Cambridge, United States; [8]Barrier Immunity Section, Lab of Viral Diseases, National Institute of Allergy and Infectious Diseases, National Institutes of Health, Cambridge, United States; [9]Immunology Section, Liver Diseases Branch, National Institute of Diabetes and Digestive and Kidney Diseases, National Institutes of Health, Bethesda, United States; [10]Department of Pathology, University of Cambridge, Cambridge, United Kingdom; [11]Mililani Mauka Elementary, Mililani, United States; [12]Department of Pathology (Neuropathology), Johns Hopkins University School of Medicine, Baltimore, United States; [13]Laboratory of Behavioral Neuroscience, National Institute on Aging, National Institutes of Health, Baltimore, United States; [14]Department of Microbiology, Ohio State University, Columbus, United States; [15]Civil Environmental and Geodetic Engineering, Ohio State University, Columbus, United States; [16]The Biodesign Center of Fundamental and Applied Microbiomics, School of Life Sciences, Center for Evolution and Medicine, Arizona State University, Tempe, United States; [17]Structural Biology Research Unit, Department of Clinical Laboratory Sciences, University of Cape Town, Rondebosch, South Africa; [18]Viral Information Institute and Department of Biology, San Diego State University, San Diego, United States

**\*For correspondence:**
buckc@mail.nih.gov

**Competing interests:** The authors declare that no competing interests exist.

**Abstract** Although millions of distinct virus species likely exist, only approximately 9000 are catalogued in GenBank's RefSeq database. We selectively enriched for the genomes of circular DNA viruses in over 70 animal samples, ranging from nematodes to human tissue specimens. A bioinformatics pipeline, Cenote-Taker, was developed to automatically annotate over 2500

**eLife digest** When scientists hunt for new DNA sequences, sometimes they get a lot more than they bargained for. Such is the case in metagenomic surveys, which analyze not just DNA of a particular organism, but all the DNA in an environment at large. A vexing problem with these surveys is the overwhelming number of DNA sequences detected that are so different from any known microbe that they cannot be classified using traditional approaches. However, some of these "known unknowns" are undoubtedly viral sequences, because only a fraction of the enormous diversity of viruses has been characterized.

This "viral dark matter" is a major obstacle for those studying viruses. This led Tisza et al. to attempt to classify some of the unknown viral sequences in their metagenomic surveys. The search, which specifically focused on viruses with circular DNA genomes, detected over 2,500 circular viral genomes. Intensive analysis revealed that many of these genomes had similar makeup to previously discovered viruses, but hundreds of them were totally different from any known virus, based on typical methods of comparison.

Computational analysis of genes that were conserved among some of these brand-new circular sequences often revealed virus-like features. Experiments on a few of these genes showed that they encoded proteins capable of forming particles reminiscent of characteristic viral shells, implying that these new sequences are indeed viruses.

Tisza et al. have added the 2,500 newly characterized viral sequences to the publicly accessible GenBank database, and the sequences are being considered for the more authoritative RefSeq database, which currently contains around 9,000 complete viral genomes. The expanded databases will hopefully now better equip scientists to explore the enormous diversity of viruses and help medics and veterinarians to detect disease-causing viruses in humans and other animals.

complete genomes in a GenBank-compliant format. The new genomes belong to dozens of established and emerging viral families. Some appear to be the result of previously undescribed recombination events between ssDNA and ssRNA viruses. In addition, hundreds of circular DNA elements that do not encode any discernable similarities to previously characterized sequences were identified. To characterize these 'dark matter' sequences, we used an artificial neural network to identify candidate viral capsid proteins, several of which formed virus-like particles when expressed in culture. These data further the understanding of viral sequence diversity and allow for high throughput documentation of the virosphere.

## Introduction

There has been a rush to utilize massive parallel sequencing approaches to better understand the complex microbial communities associated with humans and other animals. Although the bacterial populations in these surveys have become increasingly recognizable (*Lloyd-Price et al., 2017*), a substantial fraction of the reads and de novo assembled contigs in many metagenomics efforts are binned as genetic 'dark matter,' with no recognizable similarity to characterized sequences (*Krishnamurthy and Wang, 2017*; *Oh et al., 2014*). Some of this dark matter undoubtedly consists of viral sequences, which have remained poorly characterized due to their enormous diversity (*Simmonds et al., 2017*; *Paez-Espino et al., 2016*; *Emerson et al., 2018*). Recent efforts have shown that our understanding of viral diversity, even of viruses known to directly infect humans, has been incomplete (*Pastrana et al., 2018*; *Turnbaugh et al., 2007*; *Gilbert et al., 2010*). To increase the power of future studies seeking to more comprehensively catalog the virome and find additional associations between viruses and disease, reference genomes for all clades of the virosphere need be identified, annotated, and made publicly accessible.

Virus discovery has typically proven to be more difficult than discovery of cellular organisms. Whereas all known cellular organisms encode conserved sequences (such as ribosomal RNA genes) that can readily be identified through sequence analysis, viruses, as a whole, do not have any universally conserved sequence components (*O'Leary et al., 2016*; *Brister et al., 2015*; *Sullivan, 2015*; *Rohwer and Edwards, 2002*). Nevertheless, some success has been achieved in RNA virus discovery

by probing for the conserved sequences of their distinctive RNA-dependent RNA polymerase or reverse transcriptase genes in metatranscriptomic data (*Shi et al., 2016*). Also, many bacteriophages of the order *Caudovirales*, such as the families *Siphoviridae*, *Podoviridae*, and *Myoviridae*, have been reported in high numbers due to their and their hosts' culturability and their detectability using viral plaque assays (*Pope et al., 2015*; *Grose and Casjens, 2014*; *Grose et al., 2014*). The relatively abundant representation of these families in databases has allowed new variants to be recognized by high-throughput virus classification tools like VirSorter (*Roux et al., 2015*; *Gregory et al., 2019*; *Roux et al., 2019b*). In contrast, many small DNA viruses are not easily cultured (*Bedell et al., 1991*), use diverse genome replication strategies, and typically lack DNA polymerase genes such as those in large DNA viruses (*Koonin et al., 2015*). An additional challenge is that small DNA viruses with segmented genomes may have segments that do not encode recognizable homologs of known viral genes. Therefore, small DNA viruses are more sparsely represented in reference databases. However, some groups have been successful in discovery of small DNA genomes in a wide range of viromes (*Blinkova et al., 2010*; *Pastrana et al., 2018*; *Dayaram et al., 2015*; *Dayaram et al., 2016*; *Labonté and Suttle, 2013*; *Rosario et al., 2018*; *Victoria et al., 2009*).

Despite the apparent challenges in detecting small DNA viruses, many have physical properties that can be leveraged to facilitate their discovery. In contrast to the nuclear genomes of animals, many DNA virus genomes have circular topology, which allows selective enrichment through rolling circle amplification (RCA) methods (*Kim et al., 2008*). Further, the unique ability of viral capsids to protect nucleic acids from nuclease digestion and to mediate the migration of the viral genome through ultracentrifugation gradients or size exclusion columns allows physical isolation of viral genomes.

The current study grew out of an effort to find papillomaviruses (small circular DNA viruses) in humans and economically important or evolutionarily informative animals (*Pastrana et al., 2018*; *Peretti et al., 2015*). The sampling included several types of animals that might serve as laboratory models (e.g., mice, fruit flies, soil nematodes). A number of papillomaviruses were detected among a vastly larger set of circular DNA sequences that were not easily identifiable in standard BLASTN searches. The goal of the present study is to catalog and annotate the circular DNA virome from these animal tissues to understand the diversity and evolution of viral sequences. We developed a comprehensive bioinformatics pipeline, Cenote-Taker, to classify and annotate over 2500 candidate viral genomes and generate GenBank-compliant output files. Cenote-Taker is available for free public use with a graphical user interface at http://www.cyverse.org/discovery-environment.

## Results

### Virion enrichment, genome sequencing, and annotation

We have previously developed methods for discovery of new polyomavirus and papillomavirus species in skin swabs and complex tissue specimens (*Peretti et al., 2015*). Nuclease-resistant DNA from purified virions was amplified by random-primed rolling circle amplification (RCA) and subjected to deep-sequencing. Reads were de novo assembled into contigs and analyzed with a bioinformatics pipeline, Cenote-Taker (a portmanteau of *cenote*, a naturally occurring circular water pool, and *note-taker*), to identify and annotate de novo-assembled contigs with terminal direct repeats consistent with circular DNA molecules. In this pipeline, putative-closed circular sequences of greater than 1000 nucleotides (nt) were queried against GenBank's nucleotide database using BLASTN to remove circles with extensive nucleotide identity (>90% across any 500 nt window) to known sequences. Sequences with >90% identity to previously reported viral sequences represented less than 1.5% of circular contigs and are not included in further analysis. Approximate taxonomy was determined by BLASTX to a protein database derived from RefSeq virus proteins and GenBank plasmid proteins (only hits better than $1 \times 10^{-5}$ were considered). Open reading frames (ORFs) from remaining unidentified circular DNA sequences > 240 nucleotides (nt) in length were translated and used for RPS-BLAST queries of GenBank's Conserved Domain Database (CDD). ORFs that did not yield E values better than $1 \times 10^{-4}$ in RPS-BLAST were subjected to BLASTP searches of viral sequences in GenBank's nr database (*Altschul et al., 1990*; *Marchler-Bauer and Bryant, 2004*; *Marchler-Bauer et al., 2015*). For ORFs that were not confidently identified in BLAST searches, HHBlits (*Remmert et al., 2012*) was used to search the CDD, Pfam (*El-Gebali et al., 2019*), Uniprot

(*UniProt Consortium, 2019*), Scop (*Chandonia et al., 2019*), and PDB (*Burley et al., 2017*) databases. The results were used to annotate and name each sequence in a human-readable genome map as well as a format suitable for submission to GenBank. After checking the Cenote-Taker output of each genome, minor revisions were made, as needed, and files were submitted to GenBank (Bio-Project Accessions PRJNA393166 and PRJNA396064). All annotations meet or exceed recently proposed standards for uncultivated virus genomes (*Roux et al., 2019a*). Plasmid sequences were frequently detected and were discarded. Circular sequences were considered to be plasmid-like if they: 1) had a best BLASTX hit to a plasmid and 2) had no detectable virion structural genes.

Viral enrichment of the analyzed samples (based on ViromeQC [*Zolfo et al., 2019*], with alignment to prokaryotic single-copy housekeeping genes) was typically high (*Supplementary file 1*). However, even in the samples where enrichment was low, quality viral genomes could still be identified based on the bioinformatic analyses.

## Discovery of 2514 DNA viruses in animal metagenomes

Of the novel circular sequences detected in the survey, 1844 encode genes with similarity to proteins of ssDNA viruses and 55 encode genes with similarity to dsDNA viral proteins (*Figure 1A*). The large majority of genomes from this study are highly divergent from RefSeq entries (*Figure 1—figure supplement 1*). We discovered 868 genomes that had similarity to unclassified eukaryotic viruses known as circular replication-associated protein (Rep)-encoding single-stranded DNA (CRESS) viruses. The group is defined by the presence of a characteristic rolling circle endonuclease/superfamily three helicase gene (Rep) (*Zhao et al., 2019*; *Kazlauskas et al., 2019*), but has not been assigned to families by the ICTV or RefSeq. We estimate that 199 non-redundant unclassified CRESS virus genomes had been previously deposited in GenBank, and 85 are curated in RefSeq (*Figure 1B*). Also abundant was the viral family *Microviridae*, a class of small bacteriophages, with 670 complete genomes. This represents a substantial expansion beyond the 459 non-redundant microvirus genomes previously listed in GenBank (of which 44 were curated in the RefSeq database). Other genomes that were uncovered represent *Anelloviridae* (n = 170), *Inoviridae* (n = 70), *Genomoviridae* (n = 58), *Siphoviridae* (n = 18), unclassified phage (n = 14), *Podoviridae* (n = 10), *Myoviridae* (n = 7) unclassified virus (n = 6), *Papillomaviridae* (n = 4), *Circoviridae* (n = 3), unclassified *Caudovirales* (n = 3), *Bacilladnaviridae* (n = 2), *Smacoviridae* (n = 2), and *CrAssphage-like* (n = 2) (*Figure 1B*, *Supplementary file 2*). Viral families were found in association with 23 different animal species (*Figure 1C*). It was not surprising to find bacterial viruses, as all animals are presumed to have microbial communities and our sampling included tissues where these communities reside.

It is difficult to assign a host to most of the viruses from this study due to their divergence from known viral sequences. However, we searched the CRISPR database at (https://crispr.i2bc.paris-saclay.fr/crispr/BLAST/CRISPRsBlast.php), and three viruses had exact matches to CRISPR spacers in bacterial genomes (Siphoviridae sp. ctcj11:Shewanella sp. W3-18-1, Inoviridae sp. ctce6:Shewanella baltica OS195, Microviridae sp. ctbe523:Paludibacter propionicigenes WB4) and one virus had an exact match to the CRISPR spacer of an archaeon (Caudovirales sp. cthg227:Methanobrevibacter sp. AbM4), implying that these organisms are infected by these viruses. Further, the 142 anelloviruses found in human blood samples (*Supplementary file 2*) are almost certain to be bona fide human viruses based on their relatedness to known human anelloviruses.

In addition to circular genomes with recognizable similarity to known viruses, 609 circular contigs appeared to represent elements that lacked discernable similarity to known viruses (*Figure 1A,C*).

The vast majority of the de novo assembled circular genomes were <10 kb in length (*Figure 1—figure supplement 2*). This is largely due to the fact that large genomes are typically more difficult to de novo assemble from short reads. Despite these technical obstacles, our detection of a new tailed bacteriophage with a 419 kb genome (Myoviridae sp. isolate ctbc_4, GenBank Accession: MH622943), along with 45 other >10 kb circular sequences (*Figure 1—figure supplement 2*), indicates that the methods used for the current work can detect large viral genomes.

There has been a recent renewal of interest in the hypothesis that viruses may be etiologically associated with degenerative brain diseases, such as Alzheimer's disease (*Itzhaki et al., 2016*; *Eimer et al., 2018*). Conflicting literature suggests the possible presence of papillomaviruses in human brain tissue (*Coras et al., 2015*; *Chen et al., 2012*). Samples of brain tissue from individuals who died of Alzheimer's disease (n = 6) and other forms of dementia (n = 6) were subjected to virion enrichment and deep sequencing. Although complete or partial genomes of known

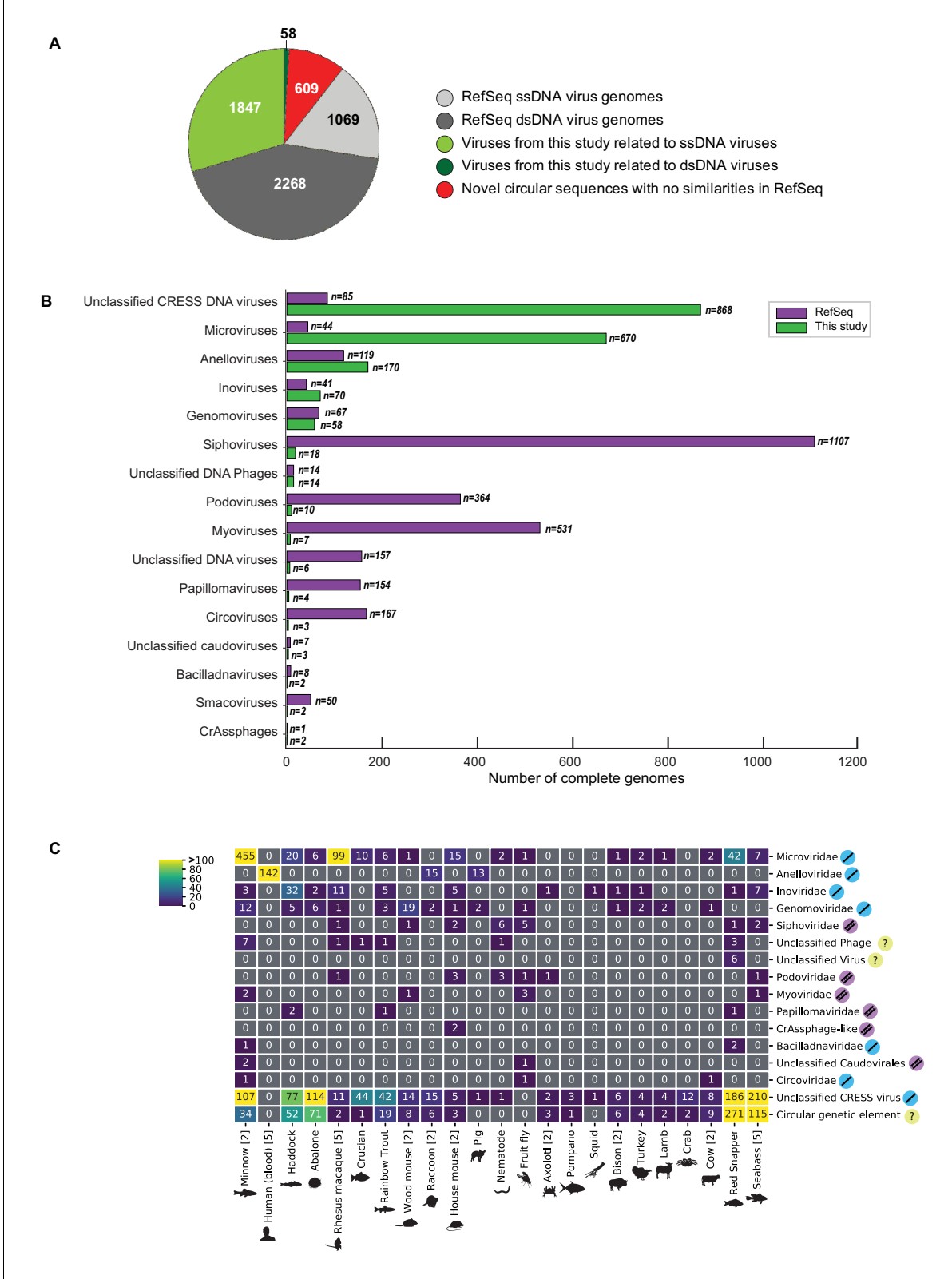

**Figure 1.** Novel viruses associated with animal samples. Gross characterization of viruses discovered in this project compared to NCBI RefSeq virus database entries. (**A**) Pie chart representing the number of viral genomes in broad categories. (**B**) Bar graph showing the number of new representatives of known viral families or unclassified groups. (**C**) Heatmap reporting number of genomes found associated with each animal species. Number of samples per species in brackets. Note that genomes in this study were assigned taxonomy based on at least one region with a BLASTX hit with an E

*Figure 1 continued on next page*

*Figure 1 continued*

value <1 × 10$^{-5}$, suggesting commonality with a known viral family. Some genomes may ultimately be characterized as being basal to the assigned family.

The online version of this article includes the following figure supplement(s) for figure 1:

**Figure supplement 1.** Divergence of proteins encoded by circular contigs.
**Figure supplement 2.** Size distribution of circular DNA sequences from this study.
**Figure supplement 3.** Mapping reads to complete viral genome references.

---

papillomaviruses, Merkel cell polyomavirus, and/or anelloviruses were observed in some samples (*Supplementary file 3*), no novel complete viral genomes were recovered (*Supplementary file 2*). No viral sequences were detected in a follow-up RNA deep sequencing analysis of the brain samples. It is difficult to know how to interpret these negative data. It is conceivable that the known viral DNA sequences observed in the Optiprep-RCA samples represent virions from blood vessels or environmental sources.

It has recently become apparent that certain nucleic acid extraction reagents are contaminated with viral nucleic acids (*Asplund et al., 2019*). To ensure we were not merely reporting the sequences of the 'reagent virome,' we performed our wet bench and bioinformatic pipeline on three independent replicates of reagent-only samples. We found no evidence of sequences of any viruses reported here or elsewhere. Further, cross-sample comparison of contigs showed that almost no sequences were found in different animal samples, aside from technical replicates. In total, six viral genomes were observed in multiple unrelated samples from at least two sequencing runs (*Supplementary file 4*). It is unclear whether this small minority of genomes (0.24% of the genomes reported in the current study) represent reagent contamination, lab contamination, or actual presence of the sequences in different types of samples.

Given the stringent requirements for sequences to be considered as belonging to a complete viral genome, as well as the largely unexplored nucleotide space of the virome, it is unsurprising that, in most samples, most reads did not align to the genomes reported in this study or virus genomes from RefSeq (*Figure 1—figure supplement 3*) (*Supplementary file 5*).

## Assignment of hallmark genes to networks shows expansion of virus sequence space

Single stranded DNA viruses, in general, have vital genes encoding proteins that mediate genome replication, provide virion structure, and, in some cases, facilitate packaging of viral nucleic acid into the virion. Being structurally conserved, these genes also tend to be important for evolutionary comparisons and can serve as important 'hallmark genes' for virus discovery and characterization. However, even structurally conserved proteins sometimes do not have enough sequence conservation as to be amenable to high confidence BLASTP searches. We therefore set out to catalog hallmark ssDNA virus genes based using protein structural prediction. Structures of hallmark genes of exemplar isolates from most established ssDNA virus families have been solved and deposited in publicly available databases such as PDB (Protein Data Bank) (*Burley et al., 2017*). Using bioinformatic tools, such as HHpred, one can assign structural matches for a given gene based on the predicted potential folds of a given amino acid sequence. HHpred has been extensively tested and validated for computational structural modeling by the structural biology community (*Meier and Söding, 2015*; *Huang et al., 2014*). The method proves especially useful for protein sequences from highly divergent viral genomes that have little similarity to annotated sequences in current databases.

We extracted protein sequences from our dataset and compiled nonredundant proteins from circular ssDNA viruses in GenBank and used them as queries in HHpred searches against the PDB, PFam, and CDD databases. We then grouped structurally identifiable sequences into hallmark gene categories and aligned them pairwise (each sequence was compared to all other sequences) using EFI-EST (*Gerlt et al., 2015*). The resulting sequence similarity networks (SSNs) were visualized with Cytoscape (*Su et al., 2014*), with each node representing an predicted protein sequence (*Figures 2–3*, *Figure 2—figure supplement 1*). Nodes (sequences) with significant amino acid similarity are connected with lines representing BLAST similarity scores better than a threshold E value. Sequence similarity network analyses, it has been proposed (*Iranzo et al., 2017*), represent relationships

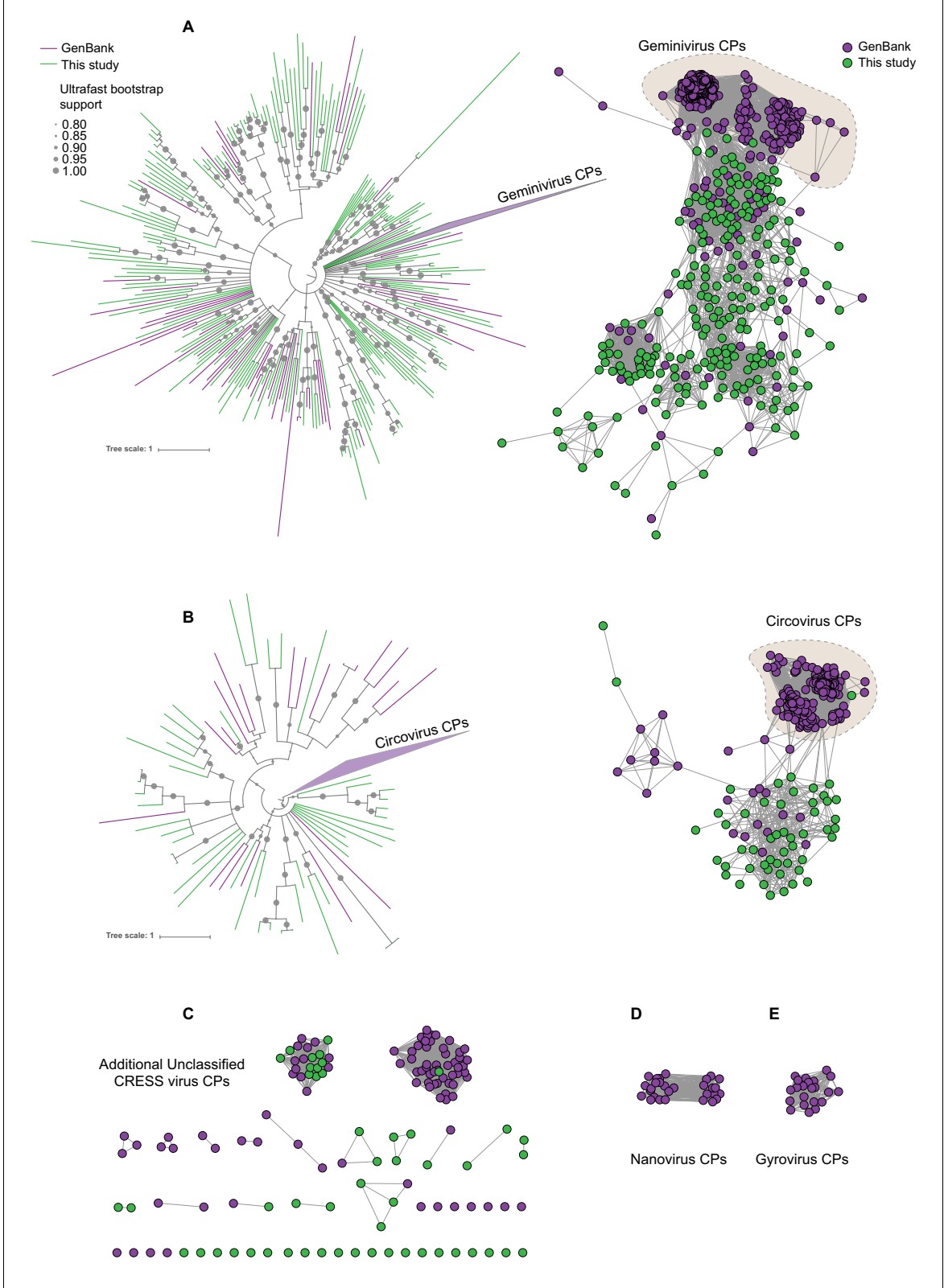

**Figure 2.** Sequence similarity network analysis of CRESS virus capsid proteins. EFI-EST was used to conduct pairwise alignments of amino acid sequences from this study and GenBank with predicted structural similarity to CRESS virus capsid proteins. The E value cutoff for the analysis was $10^{-5}$. (A) Cluster consisting of proteins with predicted structural similarity to geminivirus-like capsids and/or STNV-like capsids. The phylogenetic tree was made from all sequences in this cluster. (B) A cluster consisting of sequences with predicted structural similarity to Circovirus capsid proteins. The

*Figure 2 continued on next page*

*Figure 2 continued*

phylogenetic tree was made from all sequences in this cluster. (**C**) Assorted clusters and singletons from unclassified CRESS virus proteins that were modeled to be capsids. (**D**) Nanovirus capsids. (**E**) Gyrovirus capsids.

The online version of this article includes the following source data and figure supplement(s) for figure 2:

**Source data 1.** Phylogenetic tree file of circovirus-like capsid protein sequences, corresponding to *Figure 2*, Panel B.

**Source data 2.** Sequence similarity network of CRESS virus capsid protein sequences, corresponding to *Figure 2*.

**Source data 3.** Phylogenetic tree file of gemini- and STNV-like capsid protein sequences, corresponding to *Figure 2*, Panel A.

**Figure supplement 1.** Network Analysis of additional viral hallmark genes.

**Figure supplement 1—source data 1.** Sequence similarity network of anellovirus ORF1 protein sequences, corresponding to *Figure 2—figure supplement 1*.

**Figure supplement 1—source data 2.** Sequence similarity network of inovirus ZOT protein sequences, corresponding to *Figure 2—figure supplement 1*.

**Figure supplement 1—source data 3.** Sequence similarity network of microvirus Major Capsid protein sequences, corresponding to *Figure 2—figure supplement 1*.

**Figure supplement 1—source data 4.** Sequence similarity network of inovirus and microvirus Replication-associated protein sequences, corresponding to *Figure 2—figure supplement 1*.

**Figure supplement 2.** Phylogenetic trees of viral hallmark genes.

**Figure supplement 2—source data 1.** Phylogenetic tree file of anellovirus ORF1 protein sequences, corresponding to *Figure 2—figure supplement 2*.

**Figure supplement 2—source data 2.** Phylogenetic tree file of CRESS virus Rep protein sequences, corresponding to *Figure 2—figure supplement 2*.

**Figure supplement 2—source data 3.** Phylogenetic tree file of inovirus ZOT protein sequences, corresponding to *Figure 2—figure supplement 2*.

**Figure supplement 2—source data 4.** Phylogenetic tree file of microvirus Major Capsid protein sequences, corresponding to *Figure 2—figure supplement 2*.

**Figure supplement 2—source data 5.** Phylogenetic tree file (1 of 3) of inovirus and microvirus Replication-associated protein sequences, corresponding to *Figure 2—figure supplement 2*.

**Figure supplement 2—source data 6.** Phylogenetic tree file (2 of 3) of inovirus and microvirus Replication-associated protein sequences, corresponding to *Figure 2—figure supplement 2*.

**Figure supplement 2—source data 7.** Phylogenetic tree file (3 of 3) of inovirus and microvirus Replication-associated protein sequences, corresponding to *Figure 2—figure supplement 2*.

between viral sequences better than phylogenetic trees. Further, SSNs have previously been used for viral protein and genome cluster comparison (*Bolduc et al., 2017*; *Lima-Mendez et al., 2008*; *Lefeuvre et al., 2019*; *Kazlauskas et al., 2019*) and can be used to display related groups of viral genes in two dimensions (*Bin Jang et al., 2019*). These clusters were also used to guide the construction of meaningful phylogenetic trees (*Figure 2A–B*, *Figure 2—figure supplement 2*).

In *Figure 2*, sequences that showed a structural match to a known eukaryotic circular ssDNA virus capsid protein are displayed as a network. This general capsid type features a single beta-jellyroll fold and assembles into T = 1 virions of 20–30 nm in diameter. The network shows that sequences from this study expand and link smaller disconnected clusters of sequences found in GenBank entries (*Figure 2A–C*). Perhaps more importantly a number of previously unknown clusters were identified, providing insight into highly divergent hallmark sequences and making this capsid sequence space amenable to BLAST searches in GenBank (*Figure 2C*). Although the satellite tobacco necrosis virus (STNV) capsid protein encapsides an RNA molecule, it has previously been noted that its structure is highly similar to the capsid proteins of geminiviruses and other ssDNA viruses (*Koonin et al., 2015*; *Kraberger et al., 2015*; *Krupovic et al., 2009*; *Hipp et al., 2017*; *Bottcher et al., 2004*; *Zhang et al., 2001*) and was included as a model for populating this network.

A similar pattern can be seen in sequence similarity networks for the Rep genes of CRESS viruses (*Figure 3*). Rep genes have been the primary sequences used for taxonomy of CRESS viruses (*Zhao et al., 2019*). In this case, it was determined that a network with alignment cutoffs with E values of $1 \times 10^{-60}$ could split the data neatly into 'family-level' clusters (*Fontenele et al., 2019*; *Kraberger et al., 2019*), precisely mirroring ICTV taxonomy of CRESS viruses. Many additional family-level clusters can be discerned from unclassified CRESS viruses. Other eukaryotic and prokaryotic ssDNA virus hallmark gene networks are shown in *Figure 2—figure supplement 1*. Phylogenetic trees of networks from *Figures 2* and *3* and *Figure 2—figure supplement 1* are displayed in *Figure 2—figure supplement 2*.

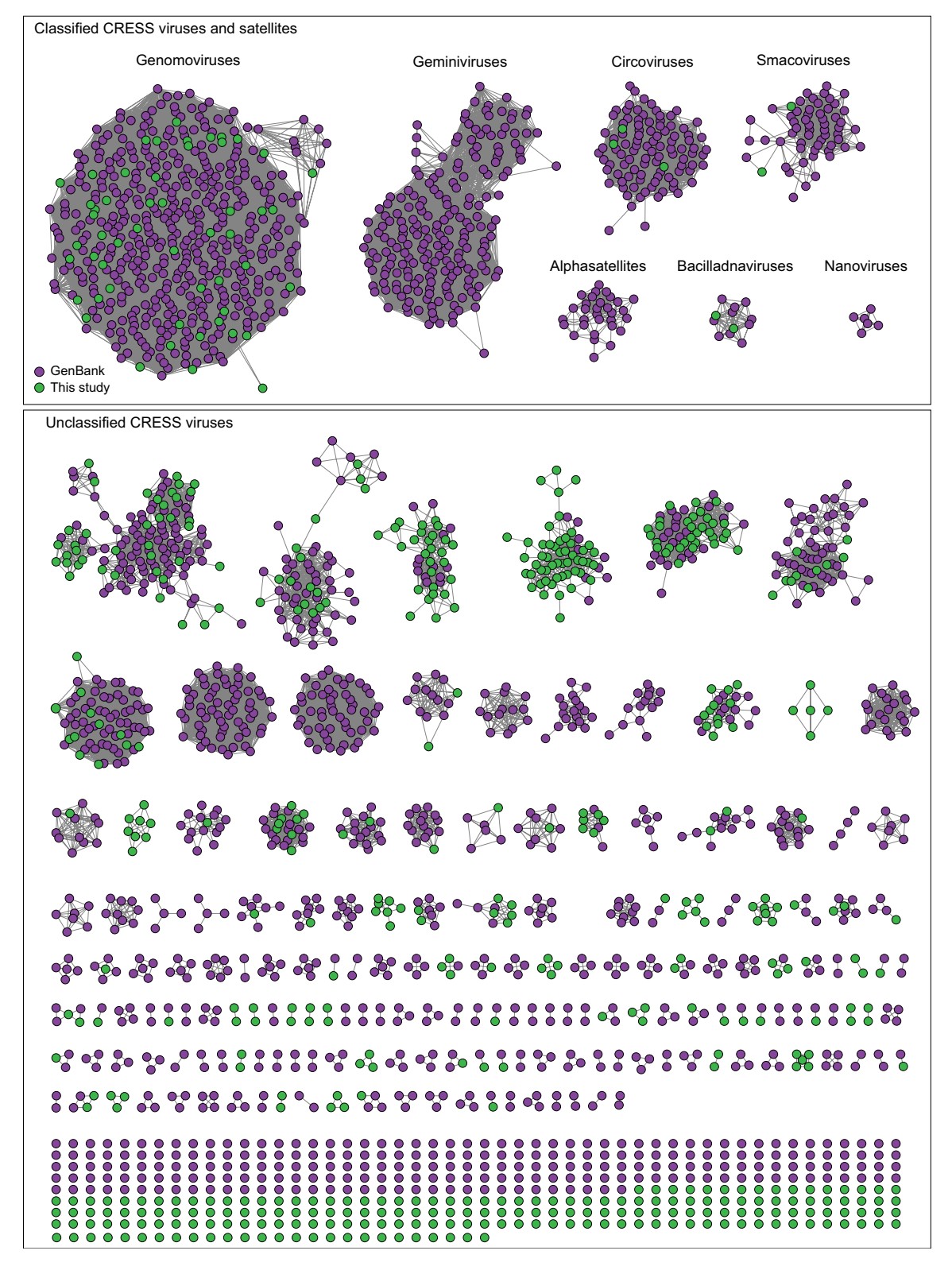

**Figure 3.** Network analysis of CRESS virus Rep proteins. EFI-EST was used to conduct pairwise alignments of amino acid sequences from this study and. GenBank that were structurally modeled to be a rolling-circle replicase (Rep). The analysis used an E value cutoff of $10^{-60}$ to divide the data into family-level clusters.

*Figure 3 continued on next page*

*Figure 3 continued*

The online version of this article includes the following source data for figure 3:

**Source data 1.** Sequence similarity network of CRESS virus Rep proteins, corresponding to *Figure 3*.

Cytoscape files of sequence similarity networks and phylogenetic trees can be found at https://ccrod.cancer.gov/confluence/display/LCOTF/DarkMatter.

## New classes of large CRESS viruses feature unconventional structural genes

Although no single family of viruses accounts for the majority of genomes in this study, these results expand the knowledge of the vast diversity of CRESS viruses, which appear to be ubiquitous among eukaryotes (*Krupovic et al., 2016*; *Zerbini et al., 2017*; *Rosario et al., 2017*; *Varsani and Krupovic, 2018*) and are likely to also infect archaea (*Díez-Villaseñor and Rodriguez-Valera, 2019*; *Kazlauskas et al., 2019*). Characterized CRESS viruses have small icosahedral virions (20–30 nm in diameter) with a simple T = 1 geometry (*Khayat et al., 2011*). This capsid architecture likely limits genome size, as nearly all previously reported CRESS virus genomes and genome segments are under 3.5 kb. Exceptions to this size rule are bacilladnaviruses, which have 4.5–6 kb genomes (*Tomaru et al., 2011*) and cruciviruses, which have 3.5–5.5 kb genomes (*Quaiser et al., 2016*). Interestingly, the genomes of these larger CRESS viruses encode capsid genes that appear to have been acquired horizontally from RNA viruses (*Kazlauskas et al., 2017*). In our dataset, eight CRESS-like circular genomes exceed 6 kb in length (*Figure 4—figure supplement 1*). Further, this study's large CRESS genomes are apparently attributable to several independent acquisitions of capsid genes from other taxa and/or capsid gene duplication events.

Notably, a large CRESS genome (CRESS virus isolate ctdh33, associated with rhabditid nematodes that were serially cultured from a soil sample) encoded three separate genes with structural homology (HHpred probability scores 97–99%) to STNV capsid (*Figure 4—figure supplement 1G*). The three predicted STNV capsid homologs in the nematode virus are highly divergent from one another, with only 28–30% amino acid similarity, but also highly divergent from other amino acid sequences in GenBank. A possible explanation for this observation is that the capsid gene array is the result of gene duplication events.

CRESS genomes ctba10, ctcc19, ctbj26, ctcd34, and ctbd1037 (ranging from 3.5 to 6.2 kb in length) also each encode two divergent capsid gene homologs (*Figure 4—figure supplement 1A,B, C,E,H*). Single genomes encoding multiple capsid genes with related but distinct amino acid sequences have been observed in RNA viruses (*Agranovsky et al., 1995*) and giant dsDNA viruses (*Schulz et al., 2017*), but we believe that this is the first time it has been reported in ssDNA viruses.

Two related large CRESS viruses (ctdb796 and ctce741) encode capsid proteins similar to those of bacilladnaviruses (*Figure 4—figure supplement 1K,M*). Interestingly, the Rep genes of the two viruses do not show close similarity to known bacilladnavirus Reps and are instead similar to the Reps of certain unclassified CRESS viruses, suggesting that CRESS ctdb796 and CRESS ctce741 are representatives of a new hybrid CRESS virus family.

Two other CRESS virus genomes (isolates ctca5 and ctgh4) encode capsid genes that show amino acid similarity to distinct groups of icosahedral T = 3 ssRNA virus capsids (*Makino et al., 2013*) (tombus- and tombus-like viruses), but not to cruciviruses or bacilladnaviruses (*Figure 4*, *Figure 4—figure supplement 1D,J*, *Figure 4—figure supplement 2A*). Further, a 6.6 kb CRESS virus (isolate ctbd466) (*Figure 4—figure supplement 1L*) was found to encode a gene with some similarity to the capsid region of the polyprotein of two newly described ssRNA viruses (ciliovirus and brinovirus (*Figure 4—figure supplement 2B*) (*Makino et al., 2013*; *Greninger and DeRisi, 2015*). Protein fold predictor Phyre[2] (*Kelley et al., 2015*) showed a top hit (58% confidence) for the capsid protein of a norovirus (ssRNA virus with T = 3 icosahedral capsid) for isolate ctbd466 (see GenBank: AXH73946).

Two CRESS genomes (ctbe30 and ctbc27) from separate Rhesus macaque stool samples combine Rep genes specific to CRESS viruses with several genes specific to inoviruses, including inovirus-like capsid genes, which encode proteins that form a filamentous virion (*Figure 4—figure supplement 1F,N*). The bacteriophage families *Inoviridae* and *Microviridae* are ssDNA viruses that replicate via the rolling circle mechanism, but they are not considered conventional CRESS viruses because they

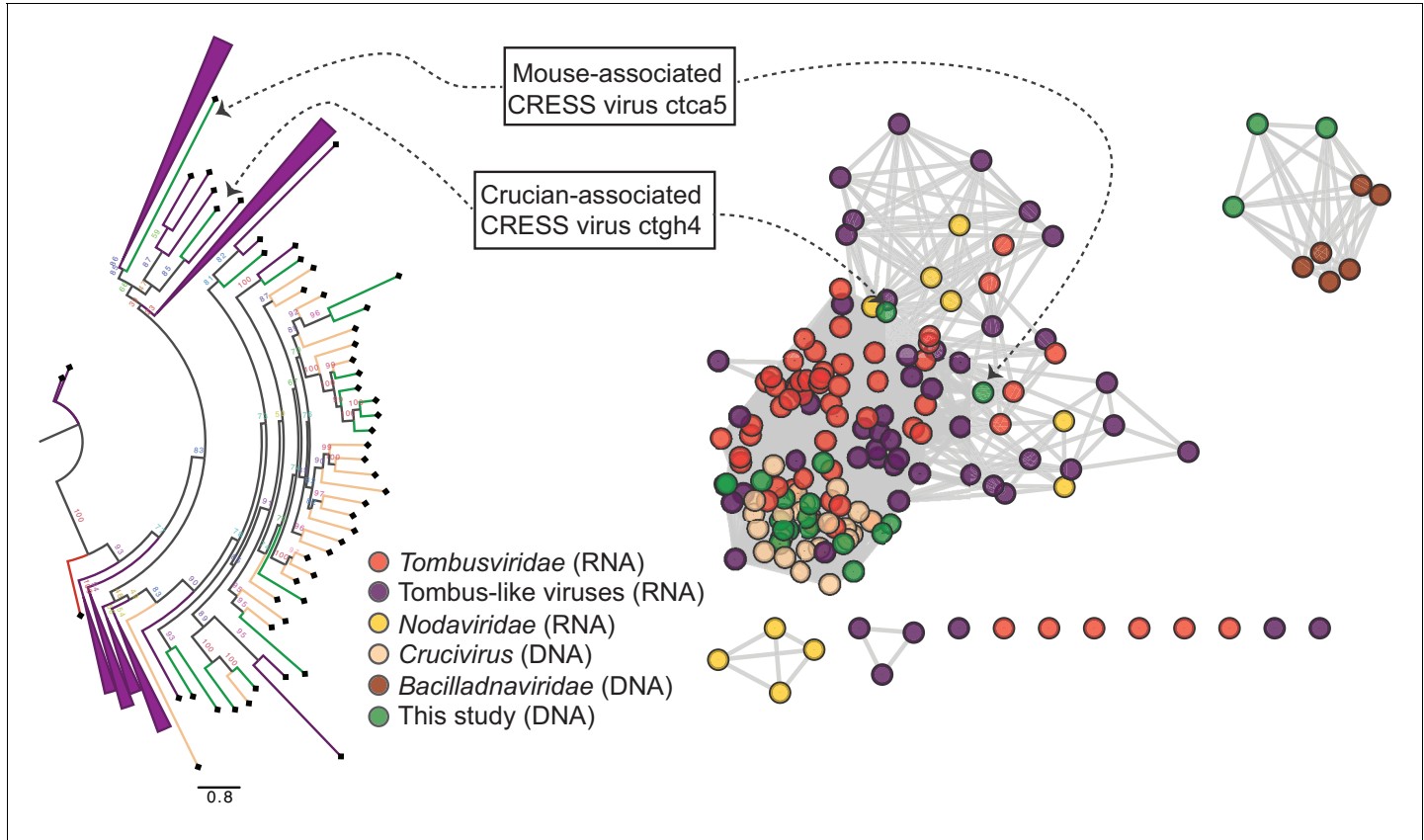

**Figure 4.** RNA virus capsid-like proteins. Sequence similarity network generated with EFI-EST (E value cutoff of $10^{-5}$) showing capsid protein sequences of select ssRNA viruses (*Nodaviridae*, *Tombusviridae*, tombus-like viruses) and ssDNA viruses (*Bacilladnaviridae* and crucivirus) together with protein sequences from DNA virus genomes observed in the present study with predicted structural similarity to an RNA virus capsid protein domain (PDB: 2IZW). Predicted capsid proteins for CRESS virus ctca5 and CRESS virus ctgh4 have no detectable similarity to any known DNA virus sequences. On the left, a phylogenetic tree representing the large cluster is displayed. Collapsed branches consist of *Tombusviridae*, tombus-like viruses, and *Nodaviridae* capsid genes.

The online version of this article includes the following source data and figure supplement(s) for figure 4:

**Source data 1.** Sequence similarity network of RNA virus-like S-domain-containing capsid protein sequences, corresponding to *Figure 4*.
**Source data 2.** Phylogenetic tree file of RNA virus-like S-domain-containing capsid protein sequences, corresponding to *Figure 4*.
**Figure supplement 1.** Genome maps of large CRESS virus genomes.
**Figure supplement 2.** Validation of proteins with predicted similarity to RNA virus capsid proteins.

exclusively infect prokaryotes and do not encode Rep genes with CRESS-like sequences. Other inovirus-like genes encoded in the ctbe30 and ctbc27 genomes include homologs of zonular occludens toxin (ZOT, a packaging ATPase) and RstB (a DNA-binding protein required for host genome integration) (*Falero et al., 2009*) (*Figure 4—figure supplement 1F,N*). TBLASTX searches using ctbe30 and ctbc27 sequences yielded large segments of similarity to various bacterial chromosomes (e.g., GenBank accession numbers AP012044 and AP018536), presumably representing integrated prophages. This suggests that ctbb30 and ctbc27 represent a previously undescribed bacteria-tropic branch of the CRESS virus supergroup.

Viral genomes discussed in this section were validated by aligning individual reads back to the contigs followed by visual inspection. No disjunctions were detected, indicating that illegitimate recombinations are not evident (see *Figure 4—figure supplement 2C* for an example).

## Network analysis of genetic 'dark matter' demonstrates conservation of gene sequence and genome structure

We defined potential viral 'dark matter' in the survey as circular contigs with no hits with E values < $1 \times 10^{-5}$ in BLASTX searches of a database of viral and plasmid proteins. We posited that leveraging

sequence similarity networks would be useful both for analyzing groups of gene homologs and for discerning which gene combinations tended to be present on related circular genomes. To categorize the 609 dark matter elements based on their predicted proteins, we used pairwise comparison with EFI-EST. A majority of translated gene sequences could be categorized into dark matter protein clusters (DMPCs) containing four or more members (*Figure 5A*). Further, groups of related dark matter elements (i.e. dark matter genome groups (DMGGs)), much like viral families, could be delineated by the presence of a conserved, group-specific marker gene. For example, DMPC1 can be thought of as the marker gene for DMGG1. Certain DMPCs tend to co-occur on the same DMGG. For instance, DMPC7 and DMPC17 ORFs are always observed in genomes with a DMPC1 ORF (i.e., DMGG1) (*Figure 5B*). This *pro tempore* categorization method is useful for visualizing the data, but we stress that is not necessarily taxonomically definitive.

HHpred, was again employed to make structural predictions for these data (*Zimmermann et al., 2018*). Instead of querying individual sequences, alignments were prepared using MAFFT (*Katoh and Standley, 2013*) for each major DMPC to identify conserved residues and increase sensitivity. Then, each alignment was used for an HHpred query. The results indicate that ten DMPCs are likely viral capsid proteins and 11 are rolling circle replicases (*Figure 5A*).

While most of the circular dark matter in the survey could be characterized using these methods, dark matter contigs represent a small remaining fraction in some samples (*Figure 5—figure supplement 1*).

## Cell culture expression of candidate 'dark matter' capsids yields particles

In contrast to viral genes such as Rep, with conserved enzymatic functions, sequences of the capsid genes are often poorly conserved, even within a given viral family (*Buck et al., 2016*). Moreover, it appears that capsid proteins have arisen repeatedly through capture and modification of different host cell proteins (*Krupovic and Koonin, 2017*). This makes it challenging to detect highly divergent capsid proteins using alignment-based approaches or even structural modeling. We therefore turned to an alignment-independent approach known as iVireons, an artificial neural network trained by comparing alignment-independent variables between a large set of known viral structural proteins and known non-structural proteins (*Seguritan et al., 2012*) (https://vdm.sdsu.edu/ivireons/). As an example of the approach, iVireons scores for DMPCs associated with DMGG1 are shown in *Figure 5C*. Other sets of iVireons scores can be seen in *Figure 5—figure supplement 2*.

Of the 17 DMGGs for which HHPRED did not identify capsid genes, iVireons predicted that ten contain at least one DMPC predicted to encode some type of virion structural protein (median score of cluster >0.70). This allowed us to generate the testable hypothesis that some of these predicted structural proteins would form virus-like particles (VLPs) if expressed in cell culture.

A subset of predicted capsid proteins were expressed in human-derived 293TT cells and/or in *E. coli* and subjected to size exclusion chromatography. Electron microscopic analysis showed that several of the predicted capsid proteins formed roughly spherical particles, whereas a negative control protein did not form particles (*Figure 6*). Although the particles were highly irregular, the DMGC11 isolate ctgh70 preparation was found to contain nuclease-resistant nucleic acids, consistent with non-specific encapsidation. The results suggest that, in multiple cases, we were able to experimentally confirm that iVireons correctly predicted the identity of viral capsid proteins.

## Discussion

Massive parallel DNA sequencing surveys characterizing microbial communities typically yield a significant fraction of reads that cannot be mapped to known genes. The present study sought to provide the research community with an expanded catalog of viruses with circular DNA genomes associated with humans and animals, as well as a means to characterize future datasets. We hope that the availability of this expanded viral sequence catalog will facilitate future investigation into associations between viral communities and disease states. Our annotation pipeline, Cenote-Taker, can be accessed via http://www.cyverse.org/discovery-environment. The CyVerse version of Cenote-Taker can readily annotate circular or linear DNA viruses. RNA viruses with polyproteins or frameshifts will require *post hoc* manual editing. Efforts could be made, for example, to apply the pipeline to previously published viromes to uncover additional viral genomes missed by other methods.

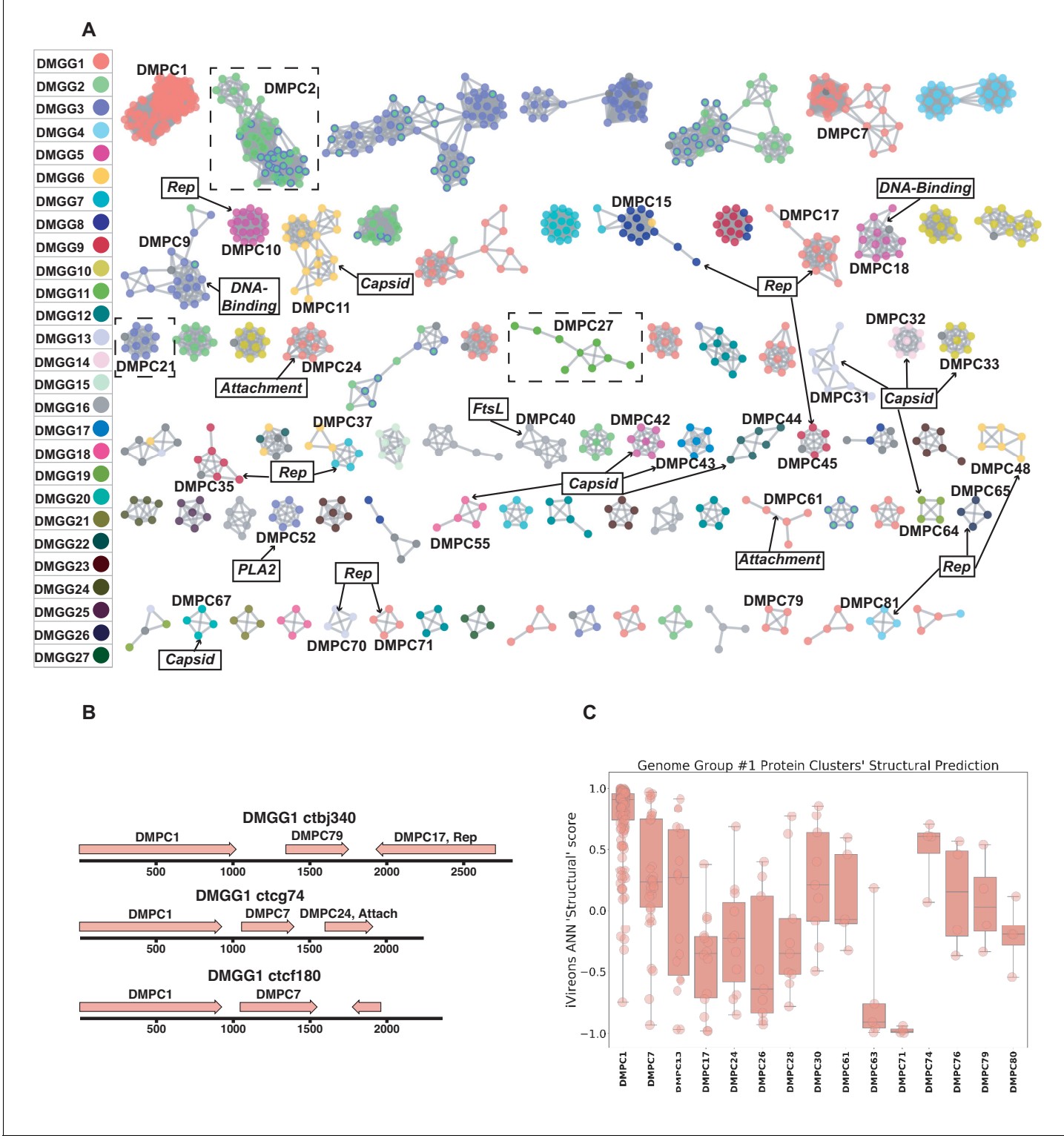

**Figure 5.** Dark matter analysis. (**A**) Sequence similarity network analysis for genes from dark matter circular sequences (minimum cluster size = 4). Clusters are colored based on assigned dark matter genome group (DMGG). Structural predictions from HHpred are indicated (>85% probability). *Rep* = rolling circle replicases typical of CRESS viruses or ssDNA plasmids. *Capsid* = single jellyroll capsid protein. *Attachment* = cell attachment proteins typical of inoviruses. *DNA-Binding* = DNA binding domain. *PLA2* = phospholipase A2. *FtsL* = FtsL like cell division protein. Clusters that contain a representative protein that was successfully expressed as a virus-like particle are outlined by a dashed rectangle (See *Figure 6*). (**B**) Maps of three examples of DMGG1 with DMPCs labeled (linearized for display). (**C**) DMGG1 iVireons 'structure' score summary by protein cluster. Scores range

*Figure 5 continued on next page*

*Figure 5 continued*

from −1 (unlikely to be a virion structural protein) to 1 (likely to be a virion structural protein). Additional iVireons score summaries can be found in *Figure 5—figure supplement 2*.

The online version of this article includes the following figure supplement(s) for figure 5:

**Figure supplement 1.** Sample characterization by iterative BLAST Searches.
**Figure supplement 2.** iVireons scores of DMGGs with candidate viral structural gene(s).

At the present time, GenBank's RefSeq database includes complete sequences for approximately 9000 viral genomes, most of which fit into 131 families recognized by the International Committee on Taxonomy of Viruses (ICTV) (*King et al., 2018*). Similarly, the IMG/VR database contains over 14,000 circular virus genomes from hundreds of studies, though some of these appear to be redundant with each other and are not comprehensively annotated (*Paez-Espino et al., 2019*). The current study, which focused on circular DNA viruses with detergent-resistant capsids, found 2514 new complete circular genomes. The availability of these comprehensively annotated genomes in GenBank contributes new information and understanding to a broad range of established, emerging, and previously unknown taxa. *Figure 3* shows dozens of potential family-level groupings within the unclassified CRESS virus supergroup. Sequences from this study contribute to 40 of such groupings and constitute the only members of seven groups. There are also 192 singleton CRESS sequences that could establish many additional family-level groups.

Although small ssDNA viruses are ubiquitous, they are often overlooked in studies that only characterize sequences that are closely related to reference genomes. In addition, ssDNA is not detected by some current DNA sequencing technologies unless second-strand synthesis (such as the RCA approach used in the current study) is conducted.

While many of the viruses discovered in this study appear to be derived from prokaryotic commensals, it is important to note that bacteriophages can contribute to human and animal diseases by transducing toxins, antimicrobial resistance proteins, or genes that alter the physiology of their bacterial hosts (*Waldor and Mekalanos, 1996*). Furthermore, interaction between animal immune systems and bacteriophages appears to be extensive (*Hodyra-Stefaniak et al., 2015*).

Over 100 distinct human anellovirus sequences were found in human blood. Anelloviruses have yet to be causally associated with any human disease, but this study indicates that we are likely still just scratching the surface of the sequence diversity of human anelloviruses. It will be important to fully catalog this family of viruses to address the field's general assumption that they are harmless.

Several of the CRESS viruses detected in this study are larger than any other CRESS virus genomes that have been described previously. In some cases, the larger size of these genomes may have been enabled by a process involving capsid gene duplication events. Further, CRESS virus acquisition of T = 3 capsids from ssRNA *Nodaviridae* and *Tombusviridae* families has been previously suggested as the origin of bacilladnaviruses (*Kazlauskas et al., 2017*) and cruciviruses (*Steel et al., 2016*; *Dayaram et al., 2016*; *Roux et al., 2013*; *Krupovic et al., 2015*), respectively. We present evidence of additional independent recombination events between CRESS viruses and ssRNA viruses and ssDNA bacteriophages. In light of these findings, it should be reiterated that only DNA (not RNA) was sequenced in our approach, so DNA/RNA in silico false recombination does not seem plausible. These data suggest that CRESS viruses are at the center of a tangled evolutionary history of viruses in which genomes change not just via gradual point mutations but also through larger scale recombination and hybridization events.

It is likely that some dark matter sequences detected in this study share a common ancestor with known viruses but are too divergent to retain discernable sequence similarity. In some cases, the dark matter circles may represent a more divergent segment of a virus with a multipartite genome. Alternatively, some of these sequences likely represent entirely new viral lineages that have not previously been recognized.

## Materials and methods

**Key resources table**

*Continued on next page*

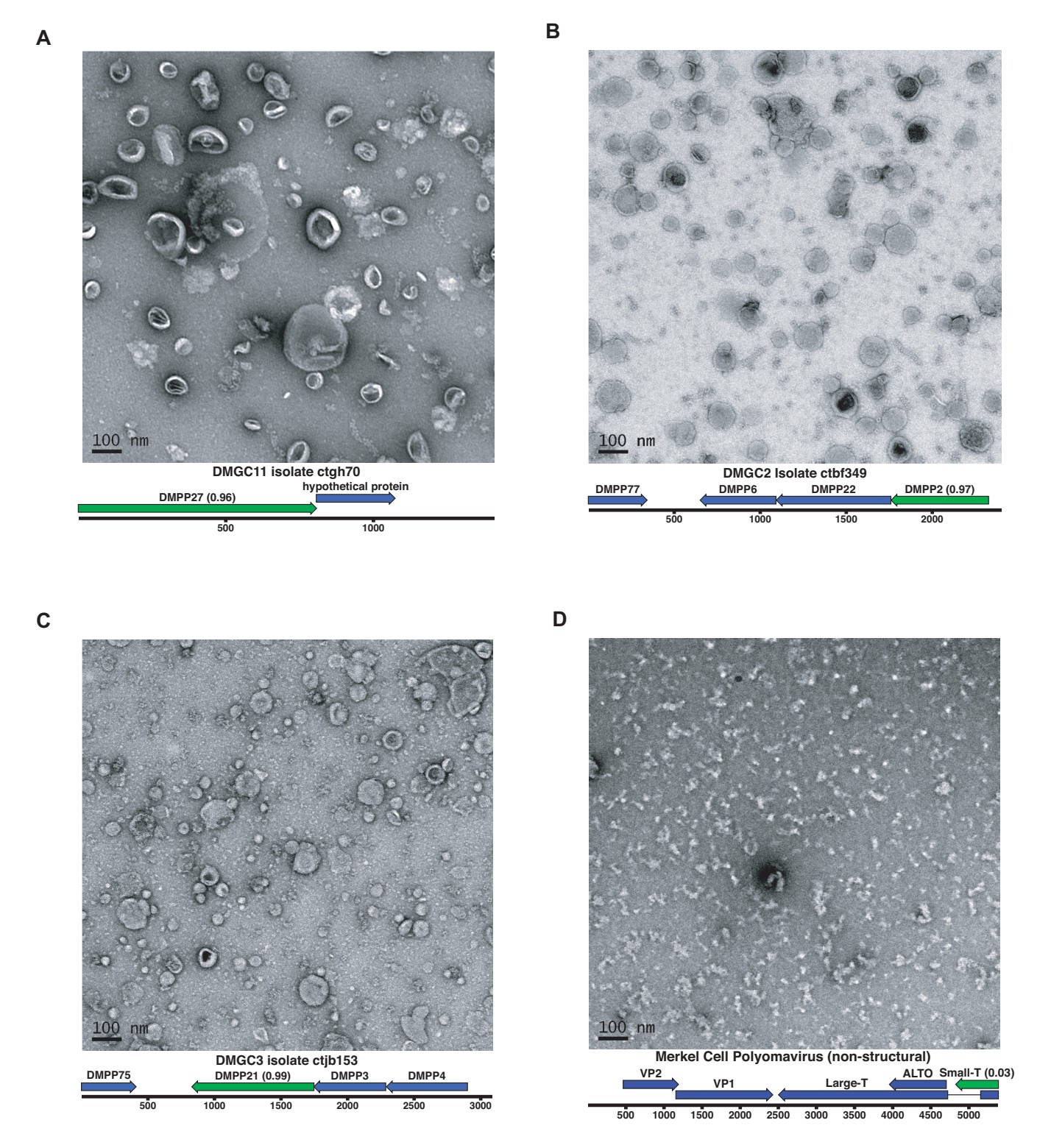

**Figure 6.** Expression of putative capsid proteins Images taken by negative stain electron microscopy. Genome maps are linearized for display purposes. Expressed genes are colored green. iVireons scores are listed in parentheses. (A-C) Images represent virus-like particles from iVireons-predicted viral structural genes. (D) Merkel cell polyomavirus small T antigen (a viral non-structural protein) is shown as a negative control.

*Continued*

| Reagent type (species) or resource | Designation | Source or reference | Identifiers | Additional information |
|---|---|---|---|---|
| Reagent type (species) or resource | Designation | Source or reference | Identifiers | Additional information |
| Strain, strain background (*Escherichia coli*) | T7 Express lysY/I<sup>q</sup> *E. coli* | NEB | Cat#: C3013I | |
| Cell line (*Homo-sapiens*) | 293TT cells | https://dtp.cancer.gov/repositories/ | NCI-293TT | Deposition to ATCC in progress |
| Recombinant DNA reagent | Dark matter capsid expression plasmids | Generated here | Lead contact | |
| Commercial assay or kit | TempliPhi 100 Amplification Kit | Sigma | Cat#: GE25-6400-10 | |
| Chemical compound, drug | Optiprep Density Medium | Sigma | Cat#: D1556-250ML | |
| Chemical compound, drug | Sepharose 4B beads | Sigma | Cat#: 4B200-100ML | |
| Software, algorithm | Cenote-Taker | http://www.cyverse.org/discovery-environment | Cenote-Taker 1.0.0 | github: https://github.com/mtisza1/Cenote-Taker |
| Software, algorithm | EFI-EST | https://efi.igb.illinois.edu/efi-est/ | EFI-EST | |
| Software, algorithm | NCBI BLAST | NCBI | RRID:SCR_004870 | |
| Software, algorithm | SPAdes assembler | http://cab.spbu.ru/software/spades/ | RRID:SCR_000131 | |
| Software, algorithm | A Perfect Circle (APC) | https://github.com/mtisza1/Cenote-Taker/blob/master/apc_ct1.pl | APC | |
| Software, algorithm | EMBOSS suite (getorf) | http://emboss.sourceforge.net/ | RRID:SCR_008493 | |
| Software, algorithm | Circlator | http://sanger-pathogens.github.io/circlator/ | RRID:SCR_016058 | |
| Software, algorithm | HHSuite | https://directory.fsf.org/wiki/Hhsuite | RRID:SCR_016133 | |
| Software, algorithm | tbl2asn | https://www.ncbi.nlm.nih.gov/genbank/tbl2asn2/ | RRID:SCR_016636 | |
| Software, algorithm | MacVector | http://macvector.com | RRID:SCR_015700 | |
| Software, algorithm | Bandage | https://rrwick.github.io/Bandage/ | Bandage | |

## Lead contact and materials availability

Further information and requests for resources and reagents should be directed to and will be fulfilled by the Lead Contact, Chris Buck (buckc@mail.nih.gov).

## Method details

### Sample collection and sequencing

De-identified human swabs and tissue specimens were collected under the approval of various Institutional Review Boards (*Supplementary file 2*). Animal tissue samples were collected under the guidance of various Animal Care and Use Committees.

Nematodes were cultured out of soil samples collected in Bethesda, Maryland, USA on OP50-Seeded NGM-lite plates (*C. elegans* kit, Carolina Biological Supply).

Viral particles were concentrated by subjecting nuclease-digested detergent-treated lysate to ultracentrifugation over an Optiprep step gradient, as previously described https://ccrod.cancer.gov/confluence/display/LCOTF/Virome (*Peretti et al., 2015*). Specifically, for each sample, no more

than 0.5 g of solid tissue was minced finely with a razorblade. Alternatively, no more than 500 µl of liquid sample was vortexed for several seconds. Samples were transferred to 1.5 ml siliconized tubes. The samples were resuspended in 500 µl Dulbecco's PBS and Triton X-100 (Sigma) detergent was added to a final concentration of 1% w/v. 1 µl of Benzonase (Sigma) was added. Samples were vortexed for several seconds. Samples were incubated in a 37°C water bath for 30 min, with brief homogenizing using a vortex every 10 min. After incubation, NaCl was added to the samples to a final concentration of 0.85M. Tubes were spun for 5 min at 5000 g. Resulting supernatants were transferred to a clean siliconized tube. Supernatant-containing tubes were spun for an additional 5 min at 5000 g. Resulting supernatants were added to iodixanol/Optiprep (Sigma) step gradients in ultracentrifuge tubes (Beckman: 326819) (equal volumes 27%, 33%, 39% iodixanol with 0.8M NaCl; total tube volume, including sample,~5.1 ml). Ultracentrifuge tubes were spun at 55,000 rpm for 3.5 hr (Beckman: Optima L-90K Ultracentrifuge). After spin, tubes were suspended over 1.5 ml siliconized collection tubes and pierced at the bottom with 25G needle. Six fractions of equal volume were collected drop-wise from each ultracentrifuge tube.

From each fraction, 200 µl was pipette to a clean siliconized tube for virus particle lysis and DNA precipitation. To disrupt virus particles, 50 µl of a 5X master mix of Tris pH 8 (Invitrogen, final conc. 50 mM), EDTA (Invitrogen, final conc. 25 mM), SDS (Invitrogen, final conc. 0.5%), Proteinase K (Invitrogen, final conc. 0.5%), DTT (Invitrogen, final conc. 10 mM) was added and mixed by pipetting up and down. Samples were heated at 50°C for 15 min. Then, proteinase K was inactivated for 10 min at 72°C. To the 250 µl of sample, 125 µl of 7.5M ammonium acetate was added and mixed by vortexing. Then, 975 µl of 95% ethanol was added and mixed by pipetting. This was incubated at room temperature for 1 hr. Then, the samples were transferred to a 4°C fridge overnight.

Samples were then restored to ambient temperature. Then, samples were spun for 1 hr at 20,000 g in a temperature-controlled tabletop centrifuge set to 21°C. Supernatant was aspirated, and 500 µl ethanol was added to each pellet. Pellets were resuspended by flicking. Then, samples were spun for 30 min at 20,000 g in a temperature-controlled tabletop centrifuge set to 21°C. Supernatant was aspirated, and samples were spun once more at 20,000 g for 3 min. Remaining liquid was carefully removed with a 10 µl micropipette. Tubes were left open and air dried for at least 10 min.

DNA from individually collected fractions of the gradient was amplified by RCA using phi29 polymerase (TempliPhi, Sigma) per manufacturer's instructions. While we expected most viral particles to travel to the middle of the gradient based on previous experiments, RCA was conducted on individual fractions spanning the gradient, in an attempt to detect viruses with different biophysical properties (*Kauffman et al., 2018*). Pooled, amplified fractions were prepared for Illumina sequencing with Nextera XT kits. Then libraries were sequenced with Illumina technology on either MiSeq or NextSeq500 sequencers. Contigs were assembled using SPAdes with the 'plasmid' setting. Circularity was confirmed by assessing assembly graphs using Bandage (*Wick et al., 2015*).

## Analysis of brain samples

Brain samples were initially analyzed by Optiprep gradient purification, RCA amplification, and deep sequencing, as described above. JC polyomavirus, which has previously been reported in brain samples (*Chalkias et al., 2018*), can display high buoyancy in Optiprep gradients (*Geoghegan et al., 2017*). Fractions from near the top of the Optiprep gradient were subjected to an alternative method of virion enrichment using microcentrifuge columns (Pierce) packed with 2 ml of Sepharose 4B Bead suspension (Sigma) exchanged into PBS. Fractions were clarified at 5000 x g for 1 min, and 200 µl of clarified extract was loaded onto the gel bed. The column was spun at 735 x g and the eluate was digested with proteinase K, ethanol-precipitated, and subjected to RCA. No additional viral sequences were detected by this method.

The brain samples were also subjected to confirmatory analysis by RNA sequencing. RNA was extracted from brain tissues with Qiagen Lipid Tissue RNeasy Mini Kit and subjected to human ribosomal RNA depletion with Thermo RiboMinus. The library was prepared with NEBNext Ultra II Directional RNA Library Prep Kit for Illumina and subjected to massive parallel sequencing on the Illumina HiSeq platform (see BioProject PRJNA513058).

## Cenote-Taker, Virus Discovery and Annotation Pipeline

Cenote-Taker, a bioinformatics pipeline written for this project and fully publicly available on CyVerse, was used for collection and detailed annotation of each circular sequence. The flow of the program can be described as follows:

1. Identifies and collects contigs (assembled with SPAdes) larger than 1000 nts
2. Predicts which contigs are circular based on overlapping ends
3. Determines whether circular contig has any ORFs of 80 AA or larger or else discards sequence
4. Uses BLASTN against GenBank 'nt' database to disregard any circular sequences that are >90% identical to known sequences across a > 500 bp window
5. Uses Circlator (*Hunt et al., 2015*) to rotate circular contigs so that a non-intragenic start codon of one of the ORFs will be the wrap point
6. Uses BLASTX against a custom virus + plasmid database (derived from GenBank 'nr' and RefSeq) to attempt to assign the circular sequence to a known family
7. Translates each ORF of 80 AA or larger
8. Uses RPS-BLAST to predict function of each ORF by aligning to known NCBI Conserved Domains
9. Generates a tbl file of RPS-BLAST results
10. Takes ORFs without RPS-BLAST hits and queries the GenBank 'nr viral' database with BLASTP
11. Generates a tbl file of BLASTP results
12. Takes ORFs without any BLASTP hits and queries HHblits (databases: uniprot20, pdb70, scop70, pfam_31, NCBI_CD)
13. Generates a tbl file of HHblits results
14. Complies with a GenBank request to remove annotations for ORFs-within-ORFs that do not contain conserved sequences
15. Combines all tbl files into a master tbl file
16. Generates a unique name for each virus based on taxonomic results
17. Generates properly formatted fsa and tbl files in a separate directory
18. Use tbl2asn to make gbf (for viewing genome maps) and sqn files (for submission to GenBank)

The source code can be found at: https://github.com/mtisza1/Cenote-Taker. (*Tisza, 2019*; copy archived at https://github.com/elifesciences-publications/Cenote-Taker).

This work utilized the computational resources of the NIH HPC Biowulf cluster. (http://hpc.nih.gov).

Genome maps were drawn, and multiple sequence alignments were computed and visualized using MacVector 16.

## Anelloviruses

Analysis of linear contigs in the survey found many instances of recognizable viral sequences. Anelloviruses are the main examples, where many contigs terminated near the GC-rich stem-loop structure that is thought to serve as the origin of replication. This segment of the anellovirus genome is presumably incompatible with the short read deep sequencing technologies used in this study. Nearly complete anellovirus genomes, defined as having a complete ORF1 gene and at least 10-fold depth of coverage, were also deposited in GenBank (*Supplementary file 2*).

## GenBank sequences

Amino Acid sequences from ssDNA viruses were downloaded in June 2018 based on categories in the NCBI taxonomy browser. As many sequences in GenBank are from identical/closely related isolates, all sequences were clustered at 95% AA ID using CD-HIT (*Fu et al., 2012*).

## Sequence Similarity Networks

Amino acid sequences from GenBank (see above) and this study were used as queries for HHsearch (the command-line iteration of HHpred) against PDB, PFam, and CDD. Sequences that had hits in these databases of 80% probability or greater were kept for further analyses. Note that capsid protein models for some known CRESS virus families have little, if any, similarity to other capsid

sequences and have not been determined (e.g. *Genomoviridae* and *Smacoviridae*) and were therefore not displayed in networks. Models used: (CRESS virus capsids network:5MJF_V, 3R0R_A, 5MJF_Ba, 4V4M_R, 4BCU_A, PF04162.11, 5J37_A, 5J09_C, 3JCI_A, cd00259, PF04660.11, PF03898.12, PF02443.14, pfam00844); (CRESS virus Rep network:4PP4_A, 4ZO0_A, 1M55_A, 1UUT_A, 1U0J_A, 1S9H_A, 4R94_A, 4KW3_B, 2HWT_A, 1L2M_A, 2HW0_A, PF08724.9, PF17530.1, PF00799.19, PF02407.15, pfam08283, PF12475.7, PF08283.10, PF01057.16, pfam00799); (*Microviridae/Inoviridae* replication-associated protein: 4CIJ_B, 4CIJ_C, PF05155.14, PF01446.16, PF11726.7, PF02486.18, PF05144.13, PF05840.12); (*Microviridae* capsid: 1M06_F, 1KVP_A, PF02305.16); (*Anelloviridae* ORF1: PF02956.13); (*Inoviridae* ZOT: 2R2A_A, PF05707.11).

## Phylogenetic trees

Sequences from this study and GenBank were grouped by structural prediction using HHpred. Then, sequences were compared by EFI-EST to generate clusters with a cut-off of $1 \times 10^{-5}$. Sequences from these clusters were then extracted and aligned with PROMALS3D (*Pei and Grishin, 2014*) using structure guidance, when possible. Structures used: (*Microviridae* MCP: 1KVP); (CRESS virus capsid STNV-like: 4V4M); (CRESS virus capsid circo-like: 3JCI); (*Inoviridae* ZOT: 2R2A); (CRESS virus Rep: 2HW0) (CRESS virus/RNA virus S Domain capsid: 2IZW). The resulting alignments were used to build trees with IQ-Tree with automatic determination of the substitution model and 1000 ultrafast bootstraps (*Nguyen et al., 2015*). Models used: (*Microviridae* MCP: Blosum62+F+G4); (*Microviridae* Rep I: Blosum62+I+G4); (*Microviridae* Rep II: LG+I+G4); (*Microviridae* Rep III: VT+I+G4); (CRESS virus/RNA virus S Domain capsid: Blosum62+F+G4); (*Circoviridae* capsid: VT+F+G4); (CRESS virus capsid STNV-like: VT+F+G4); (*Inoviridae* ZOT: VT+I+G4); (*Anelloviridae* ORF1: VT+F+G4). Trees were visualized with FigTree (http://tree.bio.ed.ac.uk/software/figtree/) and iTOL (*Letunic and Bork, 2019*).

## Expressing potential viral structural proteins in human 293TT cells

293TT cells were transfected with potential viral structural protein expression constructs for roughly 48 hr. Cells were lysed in a small volume of PBS with 0.5% Triton X-100 or Brij-58 and Benzonase (Sigma). After several hours of maturation at neutral pH, the lysate was clarified at 5000 x g for 10 min. The clarified lysate was loaded onto a 27-33–39% Optiprep gradient in PBS with 0.8 M NaCl. Gradient fractions were collected by bottom puncture of the tube and screened by PicoGreen nucleic acid stain (Invitrogen), BCA, and SDS-PAGE analysis. Electron microscopic analysis was then performed. Expression in 293TT cells of some 'dark matter' virus capsids was attempted but not successful in any case. 293TT cells were generated in-house for the previous paper (*Buck et al., 2004*), and passages from original stocks were used. Mycoplasma testing is conducted annually using MycoScope PCR Mycoplasma Detection Kit from Genlantis. Validation testing was not conducted at the time of experimentation, but the process of validation and deposition into the ATCC database is ongoing using STR profiling to authenticate human cells.

## Expressing potential viral structural proteins in *E. coli*

Several genes that were identified by iVireons as being potential viral structural proteins were cloned into plasmids with a T7 polymerase-responsive promoter. Plasmids were transfected into T7 Express lysY/Iq *E. coli*, which express T7 polymerase under the induction of IPTG. Bacteria were grown at 37˚ C in LB broth until OD600 = 0.5. Flasks were cooled to room temperature, IPTG was added to 1 mM, and cultures were shaken at room temperature for approximately 16 hr. Cells were then pelleted for immediate processing.

Total protein was extracted with a BPER (Pierce) and nuclease solution. Then, virion-sized particles were enriched from the clarified lysate using size exclusion chromatography with 2% agarose beads https://ccrod.cancer.gov/confluence/display/LCOTF/GelFiltration. Fractions were analyzed using Coomassie-stained SDS-PAGE gels for presence of a unique band corresponding to the expressed protein. Fractions of interest were analyzed using negative stain electron microscopy.

## Electron microscopy

Five µl samples were adsorbed onto a carbon-deposited copper grid for one minute. Sample was then washed 5 times on water droplets then stained with 0.5% uranyl acetate for 1 s. The negatively stained samples were examined on a FEI Tecnai T12 transmission electron microscope.

## ViromeQC

ViromeQC was run on reads from each sample corresponding to an SRA run. The 'human' setting was used, and the diamond alignment to ' 31 prokaryotic single-copy markers' was reported.

## Mapping reads to reference genomes

Viral genomes from RefSeq were downloaded from NCBI. On RefSeq and 'This study' genomes, RepeatMasker was used with '-noint' and '-hmmer' settings to mask low-complexity regions to prevent nonspecific mapping. However, this likely led to some degree of under-mapping. Reads were trimmed with fastp and aligned with Bowtie2 using default settings.

## Sequencing

Illumina sequencing was conducted at the CCR Genomics Core at the National Cancer Institute, NIH, Bethesda, MD 20892.

## Data and code availability

All reads and annotated genomes associated with this manuscript can be found on NCBI BioProject Accessions PRJNA393166 and PRJNA396064.

Cenote-Taker, the viral genome annotation pipeline, can be used by interested parties on the Cyverse infrastructure: http://www.cyverse.org/discovery-environment.

## Additional resources

Relevant protocols on lab website: https://ccrod.cancer.gov/confluence/display/LCOTF/Virome.

## Acknowledgements

This research was supported (in part)] by the Intramural Research Program of the NIH, NCI, NIDDK.

We would like to acknowledge the GenBank team at NCBI for productive discussion about their viral genome submission requirements and facilitation of annotated genome deposition.

## Additional information

### Funding

| Funder | Author |
| --- | --- |
| National Cancer Institute | Christopher B Buck |

The funders had no role in study design, data collection and interpretation, or the decision to submit the work for publication.

### Author contributions

Michael J Tisza, Conceptualization, Resources, Data curation, Software, Formal analysis, Validation, Investigation, Visualization, Methodology, Project administration; Diana V Pastrana, Conceptualization, Formal analysis, Investigation, Methodology; Nicole L Welch, Formal analysis, Investigation, Methodology; Brittany Stewart, Alberto Peretti, Yuk-Ying S Pang, Patricia A Pesavento, Formal analysis, Investigation; Gabriel J Starrett, Data curation, Software, Formal analysis; Siddharth R Krishnamurthy, Software, Formal analysis; David H McDermott, Philip M Murphy, Jessica L Whited, Bess Miller, Jason Brenchley, Stephan P Rosshart, Barbara Rehermann, John Doorbar, Olga Pletnikova, Juan C Troncoso, Susan M Resnick, Resources, Investigation; Blake A Ta'ala, Conceptualization, Resources; Ben Bolduc, Resources, Software, Investigation; Matthew B Sullivan, Conceptualization, Software; Arvind Varsani, Conceptualization, Data curation, Formal analysis, Supervision,

Investigation, Visualization, Methodology; Anca M Segall, Conceptualization, Data curation, Formal analysis, Supervision, Investigation, Methodology; Christopher B Buck, Conceptualization, Resources, Data curation, Formal analysis, Supervision, Funding acquisition, Investigation, Visualization, Methodology, Project administration

### Author ORCIDs
Diana V Pastrana ⬥ http://orcid.org/0000-0002-8084-5665
Jessica L Whited ⬥ http://orcid.org/0000-0002-3709-6515
Bess Miller ⬥ http://orcid.org/0000-0002-9868-5436
Arvind Varsani ⬥ http://orcid.org/0000-0003-4111-2415
Christopher B Buck ⬥ https://orcid.org/0000-0003-3165-8094

### Ethics
Human subjects: Patients signed informed consent forms, and protocols were approved by Institutional Review Boards at the National Cancer Institute and National Institute of Allergy and Infectious Diseases. Protocol number: 09-I-0200.

### Decision letter and Author response
Decision letter https://doi.org/10.7554/eLife.51971.sa1
Author response https://doi.org/10.7554/eLife.51971.sa2

## Additional files

### Supplementary files
- Supplementary file 1. ViromeQC enrichment scores.
- Supplementary file 2. Sample metadata and virus summary.
- Supplementary file 3. Sequences associated with brain samples.
- Supplementary file 4. Viruses observed in multiple samples.
- Supplementary file 5. Read counts for abundant viruses.
- Transparent reporting form

### Data availability
Reads, genomes, and metadata are available on NCBI (GenBank and SRA), bioprojects PRJNA393166, PRJNA396064, PRJNA513058. Code for annotating genomes is available at: https://github.com/mtisza1/Cenote-Taker (copy archived at https://github.com/elifesciences-publications/Cenote-Taker). Phylogenetic tree files and sequence similarity networks have been provided as source data and are also available at: https://ccrod.cancer.gov/confluence/display/LCOTF/DarkMatter.

The following datasets were generated:

| Author(s) | Year | Dataset title | Dataset URL | Database and Identifier |
|---|---|---|---|---|
| Tisza MJ, Buck CB | 2018 | Animal metagenomes enriched for circular DNA viruses | https://www.ncbi.nlm.nih.gov/bioproject/PRJNA393166 | NCBI BioProject, PRJNA393166 |
| Tisza MJ, Pastrana D, Buck CB | 2017 | Survey of single and double stranded DNA non-enveloped viruses on the skin of WHIM patients, healthy controls and other immune compromised patients | https://www.ncbi.nlm.nih.gov/bioproject/PRJNA396064 | NCBI BioProject, PRJNA396064 |
| Tisza MJ, Buck CB | 2019 | Metatranscriptomics of Brain Tissues from deceased dementia patients | https://www.ncbi.nlm.nih.gov/bioproject/PRJNA513058 | NCBI BioProject, PRJNA513058 |

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
