## [Decision Letter]

**Acceptance summary:**

The results of this work are of significant interest to virologists with an interest in viral diversity as well as to those interested in the association of viruses with disease. It represents a substantive advancement in the field of virus sequencing and discovery. Both the informatics and processing techniques are well described and should allow others to reproduce this work. There likely are many of these small circular viruses present in sequenced viromes, and this work provides a blueprint to allow others to identify them as well. The study also significantly increases the size of the database of known viral sequences to compare. The research and data analysis are well done and it is particularly appreciated that the authors making publicly available the software for others to use in the future. The close attention to the reviewers' comments, including the provision of additional data analysis, was much appreciated.

**Decision letter after peer review:**

Thank you for submitting your article "Discovery of several thousand highly diverse circular DNA viruses" for consideration by *eLife*. Your article has been reviewed by two peer reviewers, and the evaluation has been overseen by Karla Kirkegaard as the Senior and Reviewing Editor. The following individuals involved in review of your submission have agreed to reveal their identity: David Pride (Reviewer #2).

The reviewers have discussed the reviews with one another and the Reviewing Editor has drafted this decision to help you prepare a revised submission.

Summary:

The manuscript by Tisza et al. provides the identification and global analysis of 2514 new full-length viral genomes derived from RCA based deep sequencing of >70 animal samples with an emphasis on the small ssDNA genomes. The results of this work are of significant interest to virologists with an interest in viral diversity as well as to those interested in the association of viruses with disease. Addressing several points, however, would further improve this manuscript.

Essential revisions:

1) Please include a brief discussion as to what criteria were utilized to designate a sequence assembly as a complete genome. For example, was there a minimum/maximum size? Did the genome have to be a closed circular genome? Was it explicitly required that a genome have at least one identifiable viral associated gene? Please comment on how plasmid sequences and nucleic acids packaged within vesicles were considered/ filtered out beyond the nuclease treatment of virus preparations.

2) Please add comments on what percentage of the total sequence data (both total sequence data and unique sequence data) went into the new genomes (and into previously known viral genomes). Likewise, it would also be informative to know what percentage of sequence data was removed by the sequencing filtering criteria discussed in the manuscript.

3) Please include analysis of the overall abundance of cellular sequences present in the samples, using perhaps cellular housekeeping genes as proxies.

4) Please consider adding a table or figure that describes the distribution of the new viral genomes across the animal species sampled.

---

## [Author Response]

Essential revisions:1) Please include a brief discussion as to what criteria were utilized to designate a sequence assembly as a complete genome. For example, was there a minimum/maximum size? Did the genome have to be a closed circular genome? Was it explicitly required that a genome have at least one identifiable viral associated gene? Please comment on how plasmid sequences and nucleic acids packaged within vesicles were considered/ filtered out beyond the nuclease treatment of virus preparations.

Minimum sequence length was 1000 nucleotides and only closed circular sequences were determined to be complete (Results paragraph one). A specific exception was made for anelloviruses. While the circular sequences did not need to have similarity to known viral genes (see sections on "dark matter"), the sequences needed to have a high density of Met-initiated ORFs. We attempted to remove plasmid-like sequences by removing circular contigs that (1) had a best BLASTX hit to a plasmid and (2) had no virion structural genes (see Results subsection “Virion enrichment, genome sequencing, and annotation”). While we expect the pipeline to remove anything clearly non-viral, it remains possible that one or more of the circular "dark matter" groups could represent elements that package themselves in host-derived vesicles or virions co-opted from co-infecting viruses rather than viruses with self-encoded virion proteins.

2) Please add comments on what percentage of the total sequence data (both total sequence data and unique sequence data) went into the new genomes (and into previously known viral genomes). Likewise, it would also be informative to know what percentage of sequence data was removed by the sequencing filtering criteria discussed in the manuscript.

We added a figure (Figure 1—figure supplement 3) and a table (Supplementary file 5) to address this suggestion. Unsurprisingly, in most samples, most reads did not map to either the new genomes described here or to NCBI's Virus RefSeq database. Inspection of other contigs showed that many represent incomplete viral genomes. This underscores the difficultly of de novo assembly of complete genomes for the kind of high-quality references that this study set out to provide. We hope the reads we deposited in SRA will be fertile hunting grounds for other groups interested in viral sequence diversity.

3) Please include analysis of the overall abundance of cellular sequences present in the samples, using perhaps cellular housekeeping genes as proxies.

To answer this, we utilized the recently-published ViromeQC pipeline (Zolfo et al., 2019) to consider reads aligning to a set of housekeeping gene HMMs (Supplementary file 1). Typically only a small proportion of reads in our datasets aligned to these housekeeping genes, but there was variation sample-to-sample. This is likely because, after collection of fractions from the ultracentrifuged Optiprep gradients, each individual fraction was amplified by Phi29 polymerase in order to capture viruses of different buoyancy (Kauffman et al., 2018), potentially amplifying some bacterial genomes near the top of the gradient.

4) Please consider adding a table or figure that describes the distribution of the new viral genomes across the animal species sampled.

We've added a panel to Figure 1 (Figure 1C) detailing the number of viruses from each viral family that were found associated with each animal species as well as the 'strandedness' of each viral family. We thank the reviewers for suggesting this and other useful improvements.